

# Changes in serum amino acid levels in non-small cell lung cancer: a case-control study in Chinese population

Ke Liu[*], Jiaoyuan Li[*], Tingting Long, Yi Wang, Tongxin Yin, Jieyi Long, Ying Shen and Liming Cheng

Department of Laboratory Medicine, Tongji Hospital, Tongji Medical College, Huazhong University of Science and Technology, Wuhan, China
[*] These authors contributed equally to this work.

## ABSTRACT

**Background**. Previous studies have shown the alteration of amino acid (AA) profile in patients with non-small cell lung cancer (NSCLC). However, there is little data regarding AA profile in NSCLC in Chinese population. The aim of this study was to evaluate AA profile in Chinese NSCLC patients, explore its utility in sample classification and further discuss its related metabolic pathways.

**Methods**. The concentrations of 22 AAs in serum samples from 200 patients with NSCLC and 202 healthy controls were determined by liquid chromatography-tandem mass spectrometer (LC-MS/MS). AA levels in different tumor stages and histological types were also discussed. The performance of AA panel in classifying the cases and controls was evaluated in the training data set and validation data set based on the receiver operating characteristic (ROC) curve, and the important metabolic pathways were identified.

**Results**. The concentrations of tryptophan (Trp), phenylalanine (Phe), isoleucine (Ile), glycine (Gly), serine (Ser), aspartic acid (Asp), asparagine (Asn), cystein (Cys), glutamic acid (Glu), ornithine (Orn) and citrulline (Cit) were significantly altered in NSCLC patients compared with controls (all P-FDR < 0.05). Among these, four AAs including Asp, Cys, Glu and Orn were substantially up-regulated in NSCLC patients (FC ≥ 1.2). AA levels were significantly altered in patients with late-stage NSCLC, but not in those with early-stage when comparing with healthy controls. In terms of histological type, these AAs were altered in both adenocarcinoma and squamous cell carcinoma. For discrimination of NSCLC from controls, the area under the ROC curve (AUC) was 0.80 (95% CI [0.74–0.85]) in the training data set and 0.79 (95% CI [0.71–0.87]) in the validation data set. The AUCs for early-stage and late-stage NSCLC were 0.75 (95% CI [0.68–0.81]) and 0.86 (95% CI [0.82–0.91]), respectively. Moreover, the model showed a better performance in the classification of squamous cell carcinoma (AUC = 0.90, 95% CI [0.85–0.95]) than adenocarcinoma (AUC = 0.77, 95% CI [0.71–0.82]) from controls. Three important metabolic pathways were involved in the alteration of AA profile, including Gly, Ser and Thr metabolism; Ala, Asp and Glu metabolism; and Arg biosynthesis.

**Conclusions**. The levels of several AAs in serum were altered in Chinese NSCLC patients. These altered AAs may be utilized to classify the cases from the controls. Gly, Ser and Thr metabolism; Ala, Asp and Glu metabolism and Arg biosynthesis pathways may play roles in metabolism of the NSCLC patient.

Corresponding authors
Ying Shen, sying830@163.com
Liming Cheng, chengliming2015@163.com

## INTRODUCTION

Lung cancer is one of the most common malignant tumors which threaten human health and life. According to global cancer statistics in 2020, lung cancer is the second most common cancer in terms of incidence, and the first leading cause of death globally, accounting for approximately 18% of global cancer deaths (*Sung et al. , 2021*). Non-small cell lung cancer (NSCLC) occupies over 85% of all the lung cancers, and the most common sub-types are adenocarcinoma and squamous cell carcinoma (SCC) among them (*Goldstraw et al. , 2011*). The 5-year survival rate for localized lung cancer is 57%, while for metastatic lung cancer it is only 5% (*Siegel, Miller & Jemal, 2020*). In China, approximately 733,300 lung cancer patients occurred in 2015 according to the latest data of the national cancer registries (*Chen et al., 2016*). It is no doubt that lung cancer is still a serious public health problem in both China and the global.

As basic metabolites and regulatory factors, amino acids (AAs) play extensive effects on cancer cells (*Wei et al., 2021*). For instance, AAs account for the majority of the carbon-based biomass production, and are the dominant nitrogen source for hexosamines, nucleotides, and other nitrogenous compounds in cancer cells (*Hosios et al., 2016*; *Ma & Vosseller, 2013*; *Meng et al., 2010*; *Zetterberg & Engstrom, 1981*). AAs can also regulate gene expression and the protein phosphorylation cascade. For example, adequate essential amino acids (EAAs) stimulated MYC oncogene mRNA translation by inhibiting the GCN2-eIF2A-ATF4 stress pathway, leading to MYC-dependent transcriptional amplification (*Yue et al., 2017*). Some AAs such as glutamine (Gln), arginine (Arg) and leucine (Leu) stimulated the cell-specific phosphorylation of mTOR1, thereby regulating the conversion of intracellular proteins (*Wu, 2009*). Meanwhile, Gln, citrulline (Cit) and methionine (Met) could also activate the mTOR1 signaling pathway through modulating the concentration of phosphatidic acids (PAs) and activity of Ras homologue enriched in brain (Rheb) in the late lysosome/endosome system (LEL) (*Bourgoin-Voillard et al., 2016*). On the other hand, the metabolism of AAs is strictly regulated and closely related to other metabolic networks, such as lipid and glucose metabolism.

Although the concentrations of circulating AAs remain constant in a healthy state, numerous studies have found that various cancers, such as colorectal, lung and breast cancers, can influence the levels of AAs (*Miyagi et al., 2011*). In recent years, the AminoIndex has aroused wide attention, which facilitates the evaluation of certain health conditions and the possibility of disease by establishing multivariate indices from blood free AA profile information (*Noguchi et al., 2006*). Using these indices, researchers have discriminated various cancer types from healthy controls, including lung, colorectal, breast, gastric, prostate, gynecological and pancreatic cancers (*Miyagi et al., 2011*; *Mikami et al., 2019*). For lung cancer, AminoIndex has also been utilized for monitoring cancer treatment
(*Higashiyama et al., 2020*). These evidences suggested that AAs are suitable candidates for the study of NSCLC metabolomics, and it might be of potential value in clinical application.

At present, several studies have reported the alterations of blood free AA profile in lung cancer patients from Japan, Korea and Europe countries (*Maeda et al., 2010*; *Shingyoji et al., 2013*; *Kim et al., 2015*; *Klupczynska et al., 2016*). However, there is still little data regarding AA profile in NSCLC patients in China. Hence, a case-control study with 200 NSCLC patients and 202 healthy controls was conducted to characterize the feature of AA profile in Chinese NSCLC patients in this study. A total of 22 AAs, including eight common EAAs and 14 common non-essential amino acids (NEAAs), were determined by a liquid chromatography-tandem mass spectrometer (LC-MS/MS) method (*Choi & Coloff, 2019*; *Sivashanmugam et al., 2017*; *Papadia et al., 2018*). The alteration of AA profile in NSCLC was identified and the related metabolic pathways were discussed. We also evaluated the performance of the AA profile for the detection of NSCLC across different histological types and stages in Chinese patients.

## MATERIALS & METHODS

### Study design and participants

The case group was comprised of 200 newly diagnosed NSCLC patients who were recruited from Tongji Hospital, Tongji Medical College, Huazhong University of Science and Technology from January 2017 to May 2018. All patients were histopathologically confirmed NSCLC without radiotherapy, chemotherapy or surgery before admission. The histologic type of tumor was identified based on results of histopathological examination, and the tumor stage was confirmed according to the eighth edition of the tumor node metastasis (TNM) staging system (*Goldstraw et al., 2016*). A total of 202 healthy subjects were randomly enrolled from healthy people who underwent physical examinations in the same hospital during the same time as recruitment of patients. The healthy controls were age and gender-matched with NSCLC cases, and none of them had a history of lung cancer or other cancers. All serum samples of the participants were obtained through intravenous collection after overnight fasting and stored at −80 °C until use. This case-control study was approved by the Institutional Review Board of Tongji Medical College, Huazhong University of Science and Technology, and it does not require the informed consent of participants (IRB approval number: TJ-IRB20210806).

### Determination of free AA profile

The determination of AAs was performed on a LC-MS/MS (AB Sciex, USA) with an electrospray ionization source in positive mode. The separation was achieved on a MSLAB-45+AAC18 column (150*4.6 mm 5 µm; Beijing Mass Spectrometry Medical Research Ltd., Beijing, China) with a flow rate of 1 mL/min. The column temperature was 50 °C, and the injection volume was 5 µL.

For AAs analysis, the AAs quantitation kits (Beijing Mass Spectrometry Medical Research Ltd, Beijing, China) were used. For sample preparation, 50 µL serum was firstly mixed with 50 µL protein precipitation agent, and then vortexed for 30 s and centrifuged at 14,680 rpm for 10 mins. Then, 8 µL of the supernatant was mixed with 42 µL labeling buffer

solution which containing synthetic norvaline as internal standard. After that, 20 μL of the derivatizing solution was added for incubation at 55 °C for 15 min. After cooling down, the solution was mixed well and centrifuged at 14,680 rpm for 10 min for direct LC-MS/MS analysis. The data acquisition and quantitation were achieved by Analyst 1.6. (AB Sciex, USA).

We simultaneously detected 22 AAs: 8 EAAs including phenylalanine (Phe), tryptophan (Trp), isoleucine (Ile), leucine (Leu), methionine (Met), lysine (Lys), valine (Val), and threonine (Thr), and 14 NEAAs including glycine (Gly), serine (Ser), asparagine (Asn), aspartic acid (Asp), cystein (Cys), glutamic acid (Glu), glutamine (Gln), histidine (His), proline (Pro), arginine (Arg), tyrosine (Tyr), alanine (Ala), ornithine (Orn) and citrulline (Cit).

## Statistical analysis

Continuous variables were presented as median and interquartile range (IQR), and categorical variables were presented as counts and percentages. Due to the relatively large sample size of this study, the distribution of the data was tested by Kolmogorov–Smirnov test. Chi-square test was used to compare categorical variables, while $T$-test or Mann–Whitney $U$ test was employed to compare continuous variables in two groups. False discovery rate (FDR) was used to adjust $P$ values for multiple comparisons. An AA was identified as de-regulation if the fold change (FC) of AA level in NSCLC and control groups exceeded the threshold (FC $\geq 1.2$ or $\leq 0.8$) with $P$-FDR $< 0.05$. However, considering the essential roles of AAs in maintaining physiological homeostasis, to obtain more information and improve efficiency, all the significantly altered AAs ($P$-FDR $< 0.05$) in NSCLC patients were selected as candidates to perform pathway analysis and construct the logistic regression model for NSCLC discrimination. To ensure the credibility of the results in model construction, we randomly divided all the participants into two groups with a ratio of 7:3: 70% of the samples assigned into training set and the other 30% into validation set. Receiver operating characteristic (ROC) curve was assessed to illustrate and evaluate the performance for case classification. The MetaboAnalyst 5.0 was used for metabolic pathway analysis.

All statistical analyses were carried out using SPSS 22.0 (IBM Corporation, Armonk, NY, USA) and GraphPad Prism 8.0 (GraphPad, San Diego, CA, USA). The visualization of the model for NSCLC detection was achieved by the Deepwise & Beckman Coulter DxAI platform (https://dxonline.deepwise.com). Two-sided $P$ values of less than 0.05 were considered statistically significant.

## RESULTS

### Method validation

This LC-MS/MS method was overall well validated with linearity, precision, accuracy, analytical sensitivity and matrix effects as shown in Tables S1–S5. The linear correlation coefficients were all greater than 0.9855. The intra-day precision and inter-day precision at two levels were within 20% except that Ser, Met, Tyr and Trp were within 24% at the low level. The recovery rates were in the range of 80–120% except that Pro, His and Trp were

121%, 128% and 66% at the low level, respectively, and Cys was 128% at the high level. The limit of detection (LOD) and the limit of quantification (LOQ) were also investigated to ensure the reliable results at low concentration. A mixed experiment showed that the presence of the matrix effect didn't affect the accurate quantification of the target analytes. All these ensured the reliability of the results in this study.

## Demographics and clinical characteristics

The basic and clinical characteristics of the 200 NSCLC patients and 202 control subjects are shown in Table 1. The results showed that 51.0% of the cases were smokers, while in controls, the percentage of smokers was 40.1% ($P = 0.028$). No significant difference was found between the cases and the controls about drinking status. Besides, the average body mass index (BMI) of the cases was significantly lower than that of the controls (24.0 kg/m$^2$ *vs.* 22.8 kg/m$^2$, $P = 0.002$). Among the patients, 66.5% were diagnosed with adenocarcinoma, while 28.5% were SSC and the remaining 5% were other NSCLC. According to the eighth edition of the TNM classification system, 2.0% of the NSCLC patients were categorized as stage 0, 33.5% as stage I, 7.0% as stage II, 26.0% as stage III, 26.5% as stage IV, and the remaining 5% of patients as unclear.

In the subgroup of NSCLC, the percentage of smokers was 39.8% in the Adenocarcinoma group, while 73.7% of the SCC were smokers (Table S6, $P < 0.001$). There are more males in SCC patients than adenocarcinoma patients (89.5% *vs* 57.9%, $P < 0.001$). No significant difference was found about age, drinking status and BMI between the adenocarcinoma and SCC group. There are more male patients in late-stage than early-stage patients (77.1% *vs* 57.6%, $P = 0.004$). The percentage of smokers was 40.0% and 58.1% in early-stage and late-stage patients, respectively (Table S7, $P = 0.013$). While 17.6% and 36.2% of the early-stage and late-stage patients were drinkers, respectively ($P = 0.005$). No significant difference was found between the early-stage and the late-stage patients about age and BMI.

## Alteration of serum AA concentrations in NSCLC patients

The comparison of serum AA concentrations between the case and control groups is presented in Table 2. After multiple testing using FDR method, the concentrations of Trp, Phe, Ile, Gly, Ser, Asp, Asn, Cys, Glu, Orn, and Cit were significantly altered in serum of NSCLC patients when comparing with healthy controls (all $P$-FDR < 0.05). Among these, four AAs including Asp, Cys, Glu and Orn were substantially up-regulated (FC $\geq$ 1.2) in serum of NSCLC patients.

## AA concentrations in patients with different histological types and clinical stages

The results of subgroup analysis based on histological type are shown in Table 3. After multiple testing using FDR method, the concentrations of Trp, Gly, Ser, Asp, Asn, Cys, Glu, Gln, Orn, and Cit were significantly altered in serum of adenocarcinoma patients when comparing with healthy controls (all $P$-FDR < 0.05). Among these, three AAs including Asp, Cys and Orn were substantially up-regulated (FC $\geq$ 1.2) in serum of adenocarcinoma patients. And the concentrations of Phe, Asp, Cys, and Glu were significantly altered in

**Table 1  Demographics and clinical characteristics.**

|  |  | Cases N = 200 | Controls N = 202 | P-value |
|---|---|---|---|---|
| Gender |  |  |  | 0.568 |
|  | Males | 138 (69.0%) | 134 (66.3%) |  |
|  | Females | 62 (31.0%) | 68 (33.7%) |  |
| Age |  | 58 (34–81) | 57 (45-80) | 0.128 |
| Smoking |  |  |  | 0.028 |
|  | Yes | 102 (51.0%) | 81 (40.1%) |  |
|  | No | 98 (49.0%) | 121 (59.9%) |  |
| Drinking |  |  |  | 0.125 |
|  | Yes | 57 (28.5%) | 72 (35.6%) |  |
|  | No | 143 (71.5%) | 130 (64.4%) |  |
| BMI |  |  |  | 0.002 |
|  | <18.5 | 22 (11.0%) | 6 (3.0%) |  |
|  | 18.5–24.0 | 106 (53.0%) | 98 (48.5%) |  |
|  | 24.0–28.0 | 63 (31.5%) | 80 (39.6%) |  |
|  | ≥28.0 | 9 (4.5%) | 18 (8.9%) |  |
| Histologic type of cancer |  |  |  |  |
|  | Adenocarcinoma | 133 (66.5%) |  |  |
|  | SCC | 57 (28.5%) |  |  |
|  | Others | 10 (5.0%) |  |  |
| Tumor stage |  |  |  |  |
|  | 0 | 4 (2.0%) |  |  |
|  | I | 67 (33.5%) |  |  |
|  | II | 14 (7.0%) |  |  |
|  | III | 52 (26.0%) |  |  |
|  | IV | 53 (26.5%) |  |  |
|  | Unknown | 10 (5.0%) |  |  |

**Notes.**

Chi-square test was used to compare categorical variables. Chi-square test: (1) gender, $\chi^2 = 0.326$, $df = 1$, (2) smoking, $\chi^2 = 4.816$, $df = 1$, (3) drinking, $\chi^2 = 2.353$, $df = 1$, (4) BMI, $\chi^2 = 14.468$, $df = 3$.
$T$-test (normal distribution) was employed to compare continuous variables. $T$-test: (1) age, $F = 33.772$, $df = 400$, 95% CI $(-0.289$ to $2.289)$.

serum of SCC patients when comparing with healthy controls (all $P$-FDR < 0.05). Among these, three AAs including Asp, Cys and Glu were substantially up-regulated (FC ≥1.2) in serum of SCC patients. Besides, the concentration of Asp was significantly altered and substantially down-regulated in serum of adenocarcinoma patients when comparing with SCC patients ($P$-FDR < 0.05, FC = 0.69).

AA levels in patients with different clinical stages are shown in Fig. 1. After multiple testing using FDR method, the concentration of Glu was significantly altered in serum of early-stage NSCLC patients ($P$-FDR = 0.044), whilst the concentrations of Phe, Trp, Ile, Asp, Cys, Glu, Orn, Gly, Ser, Asn and Arg were significantly altered in serum of late-stage NSCLC patients when comparing with healthy controls (all $P$-FDR < 0.05). Four AAs including Asp, Cys, Glu and Orn were substantially up-regulated (FC ≥ 1.2) in

**Table 2  Alteration in serum amino acid concentrations in NSCLC patients.**

| Amino acid | Case $N = 200$ | | Control $N = 202$ | | Fold change | *P*-value | *P*-FDR |
|---|---|---|---|---|---|---|---|
| | Median | IQR | Median | IQR | | | |
| Phenylalanine[a] | 104.00 | 45.10 | 96.90 | 35.70 | 1.07 | 0.002 | 0.006 |
| Tryptophan[a] | 51.15 | 25.10 | 57.70 | 27.50 | 0.89 | 0.005 | 0.012 |
| Isoleucine[a] | 87.30 | 39.20 | 81.85 | 32.60 | 1.07 | 0.016 | 0.032 |
| Leucine[a] | 171.50 | 62.00 | 161.00 | 62.30 | 1.07 | 0.033 | 0.061 |
| Methionine[a] | 28.30 | 11.80 | 29.65 | 12.20 | 0.95 | 0.755 | 0.755 |
| Lysine[a] | 227.50 | 117.30 | 222.00 | 113.80 | 1.02 | 0.390 | 0.477 |
| Valine[a] | 292.50 | 112.30 | 275.00 | 94.30 | 1.06 | 0.131 | 0.206 |
| Threonine[a] | 152.00 | 74.80 | 148.00 | 63.00 | 1.03 | 0.268 | 0.369 |
| Glycine[b] | 366.00 | 175.30 | 326.00 | 138.30 | 1.12 | 0.001 | 0.004 |
| Serine[b] | 191.00 | 80.80 | 171.00 | 67.80 | 1.12 | 0.001 | 0.004 |
| Asparagine[b] | 63.05 | 25.30 | 57.70 | 24.70 | 1.09 | 0.002 | 0.006 |
| Aspartic acid[b] | 46.75 | 31.80 | 32.55 | 16.90 | 1.44 | <0.001 | <0.001 |
| Cysteine[b] | 68.05 | 49.10 | 53.75 | 44.00 | 1.27 | <0.001 | <0.001 |
| Glutamic acid[b] | 126.00 | 82.80 | 104.50 | 63.40 | 1.21 | <0.001 | <0.001 |
| Glutamine[b] | 524.00 | 221.80 | 504.50 | 238.50 | 1.04 | 0.076 | 0.129 |
| Histidine[b] | 62.45 | 33.50 | 60.05 | 36.70 | 1.04 | 0.559 | 0.647 |
| Proline[b] | 218.50 | 101.50 | 213.50 | 82.50 | 1.02 | 0.286 | 0.37 |
| Arginine[b] | 112.00 | 71.40 | 107.50 | 63.80 | 1.04 | 0.150 | 0.22 |
| Tyrosine[b] | 75.90 | 51.10 | 72.65 | 50.00 | 1.04 | 0.733 | 0.755 |
| Alanine[b] | 471.50 | 227.30 | 474.00 | 169.80 | 0.99 | 0.604 | 0.664 |
| Ornithine[b] | 113.00 | 79.30 | 92.50 | 59.10 | 1.22 | <0.001 | <0.001 |
| Citrulline[b] | 30.55 | 18.90 | 26.75 | 15.90 | 1.14 | 0.016 | 0.032 |

**Notes.**
[a] Essential amino acids.
[b] Not essential amino acids.
Mann–Whitney $U$ test was used for continuous variables that did not normally distributed.
Benjamini–Hochberg procedure was used for multiple testing correction.

serum of late-stage NSCLC patients. Although there was no significant difference in AA levels between the early-stage and late-stage NSCLC, several AA levels showed consecutive changes with the disease progression. For example, the Trp concentration gradually decreased in the control group, early-stage and late-stage NSCLC patients successively, while concentrations of AAs such as Ile, Gly, Ser, Asn, Asp, Cys, Glu and Orn showed a gradually increased trend.

## Metabolic pathway analysis of AAs in NSCLC

Metabolic pathway analysis was performed based on the altered 11 AAs between the NSCLC group and the control group. As illustrated in Fig. 2 and Table S8, we identified three significant pathways with important impact (impact factor > 0.4, *P*- FDR < 0.05), including Gly, Ser and Thr metabolism which hits three AAs including Ser, Gly, Cys (*P*-FDR = 0.016, impact factor = 0.463), Ala, Asp and Glu metabolism which hits three AAs

Liu et al. (2022), *PeerJ*, DOI 10.7717/peerj.13272

**Table 3** Subgroup analysis of amino acid concentrations in patients based on histological types.

| Amino acid | Adenocarcinoma N = 133 | | | | | SCC N = 57 | | | | | Fold change[e] | P-value[e] | P-FDR[e] |
|---|---|---|---|---|---|---|---|---|---|---|---|---|---|
| | Median | IQR | Fold change[c] | P-value[c] | P-FDR[c] | Median | IQR | Fold change[d] | P-value[d] | P-FDR[d] | | | |
| Phenylalanine[a] | 99.60 | 34.80 | 1.03 | 0.212 | 0.301 | 114.00 | 59.70 | 1.18 | <0.001 | <0.001 | 0.87 | 0.008 | 0.088 |
| Tryptophan[a] | 51.70 | 25.00 | 0.90 | 0.006 | 0.019 | 48.10 | 25.80 | 0.83 | 0.087 | 0.216 | 1.07 | 0.925 | 0.969 |
| Isoleucine[a] | 86.50 | 35.00 | 1.06 | 0.049 | 0.098 | 87.00 | 52.20 | 1.06 | 0.113 | 0.216 | 0.99 | 0.789 | 0.882 |
| Leucine[a] | 170.00 | 50.00 | 1.06 | 0.180 | 0.292 | 175.00 | 108.00 | 1.09 | 0.038 | 0.120 | 0.97 | 0.245 | 0.797 |
| Methionine[a] | 28.90 | 11.00 | 0.97 | 0.856 | 0.856 | 26.70 | 13.10 | 0.90 | 0.460 | 0.633 | 1.08 | 0.316 | 0.797 |
| Lysine[a] | 234.00 | 106.00 | 1.05 | 0.219 | 0.301 | 210.00 | 158.50 | 0.95 | 0.879 | 0.938 | 1.11 | 0.519 | 0.882 |
| Valine[a] | 281.00 | 95.50 | 1.02 | 0.285 | 0.369 | 295.00 | 149.00 | 1.07 | 0.250 | 0.394 | 0.95 | 0.759 | 0.882 |
| Threonine[a] | 154.00 | 68.00 | 1.04 | 0.065 | 0.119 | 143.00 | 86.50 | 0.97 | 0.553 | 0.715 | 1.08 | 0.103 | 0.567 |
| Glycine[b] | 378.00 | 162.50 | 1.16 | 0.001 | 0.006 | 349.00 | 224.50 | 1.07 | 0.113 | 0.216 | 1.08 | 0.582 | 0.882 |
| Serine[b] | 191.00 | 68.00 | 1.12 | 0.007 | 0.019 | 190.00 | 88.50 | 1.11 | 0.023 | 0.100 | 1.01 | 0.719 | 0.882 |
| Asparagine[b] | 63.90 | 24.40 | 1.11 | 0.005 | 0.018 | 58.80 | 35.00 | 1.02 | 0.115 | 0.216 | 1.09 | 0.802 | 0.882 |
| Aspartic acid[b] | 39.90 | 26.70 | 1.23 | <0.001 | <0.001 | 57.50 | 32.90 | 1.77 | <0.001 | <0.001 | 0.69 | <0.001 | <0.001 |
| Cysteine[b] | 67.40 | 47.80 | 1.25 | <0.001 | 0.009 | 76.10 | 47.80 | 1.42 | 0.001 | 0.006 | 0.89 | 0.280 | 0.797 |
| Glutamic acid[b] | 122.00 | 77.50 | 1.17 | <0.001 | <0.001 | 135.00 | 93.90 | 1.29 | <0.001 | <0.001 | 0.90 | 0.689 | 0.882 |
| Glutamine[b] | 528.00 | 213.50 | 1.05 | 0.011 | 0.027 | 488.00 | 287.50 | 0.97 | 0.733 | 0.896 | 1.08 | 0.042 | 0.308 |
| Histidine[b] | 65.90 | 31.60 | 1.10 | 0.186 | 0.292 | 57.80 | 45.60 | 0.96 | 0.881 | 0.938 | 1.14 | 0.311 | 0.797 |
| Proline[b] | 217.00 | 97.00 | 1.02 | 0.493 | 0.542 | 223.00 | 124.00 | 1.04 | 0.330 | 0.483 | 0.97 | 0.675 | 0.882 |
| Arginine[b] | 111.00 | 72.80 | 1.03 | 0.492 | 0.542 | 114.00 | 85.70 | 1.06 | 0.118 | 0.216 | 0.97 | 0.326 | 0.797 |
| Tyrosine[b] | 75.80 | 49.70 | 1.04 | 0.612 | 0.641 | 76.00 | 57.90 | 1.05 | 0.896 | 0.938 | 1.00 | 0.635 | 0.882 |
| Alanine[b] | 477.00 | 201.50 | 1.01 | 0.411 | 0.502 | 433.00 | 299.00 | 0.91 | 0.954 | 0.954 | 1.10 | 0.593 | 0.882 |
| Ornithine[b] | 116.00 | 76.80 | 1.25 | <0.001 | <0.001 | 110.00 | 90.20 | 1.19 | 0.031 | 0.112 | 1.05 | 0.611 | 0.882 |
| Citrulline[b] | 30.20 | 15.60 | 1.13 | 0.017 | 0.037 | 32.00 | 27.30 | 1.20 | 0.186 | 0.315 | 0.94 | 0.991 | 0.991 |

**Notes.**

[a] Essential amino acids.

[b] Not essential amino acids.

[c] Compared between adenocarcinoma and control.

[d] Compared between squamous cell carcinoma and control.

[e] Compared between adenocarcinoma and squamous cell carcinoma.

Mann–Whitney U test was used for continuous variables that did not normally distributed.

Benjamini–Hochberg procedure was used for multiple testing correction.

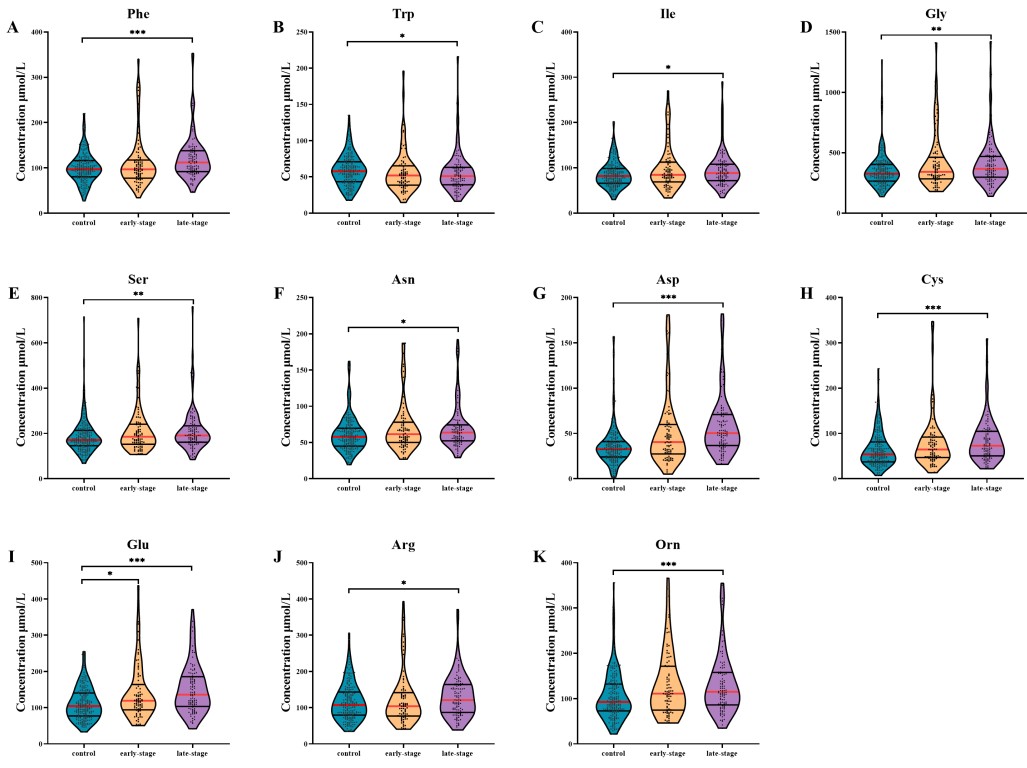

**Figure 1** **Subgroup analysis of amino acid concentrations in patients based on clinical stages.** (A–K) The concentrations of Phe (A), Trp (B), Ile (C), Gly (D), Ser (E), Asn (F), Asp (G), Cys (H), Glu (I), Arg (J), Orn (K) in patients with different stages. Mann–Whitney $U$ test was used for continuous variables that did not normally distributed. Benjamini–Hochberg procedure was used for multiple testing correction. See Supplemental Information 5 for the statistical result report.

including Glu, Asp, Asn (*P*- FDR = 0.017, impact factor=0.421), and Arg biosynthesis which hits four AAs including Glu, Asp, Cit, Orn (*P*- FDR ≤0.001, impact factor = 0.406).

## The classification of NSCLC patients and control group

The AA panel for NSCLC classification was composed of the 11 AAs which of the levels were altered in NSCLC patients compared with controls, including Phe, Trp, Ile, Gly, Ser, Asn, Asp, Cys, Glu, Orn, and Cit, and the model was generated using logistic regression. As a result, the area under the curve (AUC) of the ROC was 0.80 (95% CI [0.74–0.85]) in the training data set (Fig. 3A). In the validation data set, the AUC of the model also reached 0.79 (95% CI [0.71–0.87]) (Fig. 3A). These results demonstrated that the AA panel may be of potential value in NSCLC classification and detection.

To further investigate the performance of the model in discriminating NSCLC patients across different histological types and clinical stages, the AUCs for each histological type and stage were also estimated. For early-stage NSCLC, the AUC was 0.75 (95% CI [0.68–0.81]), while for late-stage NSCLC, the AUC reached 0.86 (95%CI [0.82–0.91]) (Fig. 3B). Moreover, the model showed large discrepancy in the detection of adenocarcinoma and

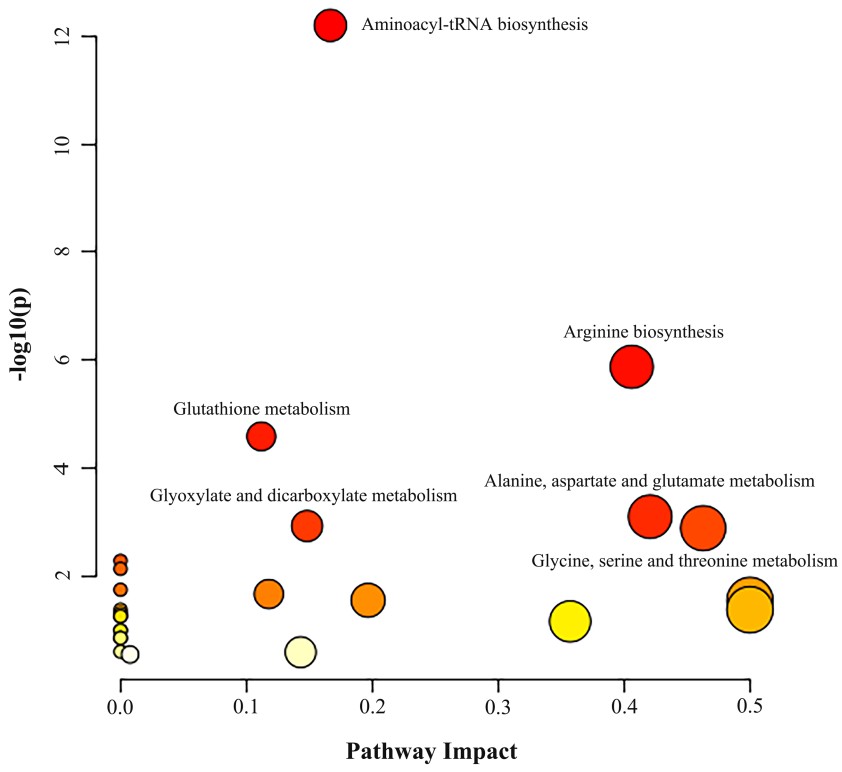

**Figure 2** Analysis of amino acid metabolic pathways in NSCLC patients. MetaboAnalyst 5.0 was used for metabolic pathway analysis. See Supplemental Information 4 for the statistical result report.

SCC, with the AUC of 0.77 (95% CI [0.71–0.82]) for adenocarcinoma, while as high as 0.90 (95%CI [0.85–0.95]) for SCC (Fig. 3B).

## DISCUSSION

Many previous reports have revealed that the metabolism was perturbed in cancer patients, including the metabolism of AAs (*Li & Zhang, 2016*; *Vander Heiden, Cantley & Thompson, 2009*). As basic building blocks of proteins, AAs play crucial roles in maintaining normal life activities and meeting the nutritional and energy requirements of tumor cells as well (*Vettore, Westbrook & Tennant, 2020*; *Tabe, Lorenzi & Konopleva, 2019*). Previous studies showed that there were substantial changes in AA profile in lung cancer, and an index derived from the altered AAs, the AminoIndex, has been utilized for lung cancer screening and cancer treatment monitoring (*Miyagi et al., 2011*; *Mikami et al., 2019*; *Higashiyama et al., 2020*; *Maeda et al., 2010*; *Kim et al., 2015*; *Klupczynska et al., 2016*; *Proenza et al., 2003*). At present, the common methods for determining the content of AAs in fluid samples include capillary electrophoresis (CE), ion-exchange chromatography (IEX), high performance liquid chromatography (HPLC), and LC-MS/MS, etc (*Smith, 1999*; *Smon et al., 2019*; *Mo et al., 2013*; *Phipps et al., 2020*). Compared with other methods, the LC-MS/MS method which adopted in our study has the advantages of high throughput, good selectivity, time-saving and cost-effective, thus has been widely used in clinical

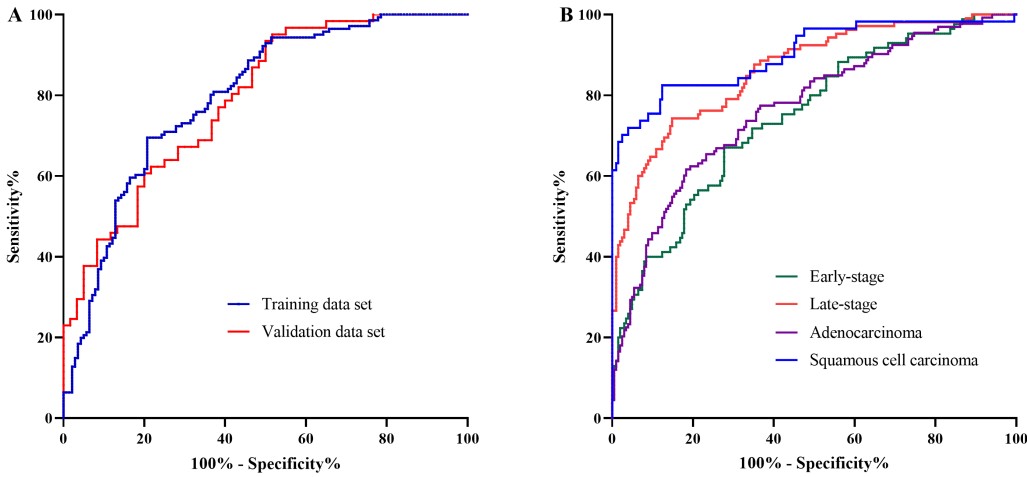

**Figure 3** **The classification of NSCLC patients and control groups.** The amino acid panel including 11 differential amino acids (Phe, Trp, Ile, Gly, Ser, Asn, Asp, Cys, Glu, Orn, Cit). (A) The ROC curve for the training data set and the validation data set. The training data set: AUC = 0.80, 95%CI [0.74−0.85]), SE = 0.03, $P < 0.001$. The validation data set: AUC = 0.79, 95%CI [0.71−0.87], SE = 0.04, $P < 0.001$. (B) The ROC curve for NSCLC patients with different stages and histological types. Early-stage NSCLC: AUC = 0.75, 95%CI [0.68−0.81], SE = 0.03, $P < 0.001$. Late-stage NSCLC: AUC = 0.86, 95%CI [0.82−0.91], SE = 0.02, $P < 0.001$. Adenocarcinoma: AUC = 0.77, 95%CI [0.71−0.82], SE = 0.03, $P < 0.001$. SCC: AUC = 0.90, 95%CI [0.85−0.95], SE = 0.03, $P < 0.001$.

assessment of AAs. Based on the application of amino derivative markers, this method has greatly improved the accuracy, precision, and detection range of AAs determination (*Le et al., 2014*). In this study, we found that the levels of 11 out of 22 AAs were significantly altered between the NSCLC and the healthy people in Chinese population. Three important metabolic pathways were involved in the metabolism of these AAs, including glycine, serine and threonine metabolism; alanine, aspartate and glutamate metabolism; and arginine biosynthesis. In addition, this AA panel were of potential value in sample classification of NSCLC.

The current study identified 11 altered AAs in serum of NSCLC, included Phe, Trp, Ile, Gly, Ser, Asp, Asn, Cys, Glu, Orn, and Cit. In previous studies, circulating Phe, Ile, Ser, Asp and Orn were found to be elevated while Trp showed a downward trend in lung cancer, which coincided with our results (*Miyagi et al., 2011*; *Maeda et al., 2010*; *Shingyoji et al., 2013*; *Kim et al., 2015*; *Klupczynska et al., 2016*; *Proenza et al., 2003*; *Zhao et al., 2014*). However, some inconsistent results were also found in several studies. For instance, Zhao's team and Kim's group (*Kim et al., 2015*) reported down-regulated Gly concentration, while Maeda's team, Miyagi's group (*Miyagi et al., 2011*) and our study observed that Gly was up-regulated in NSCLC patients. Remarkably, abnormal levels of these AAs have been found in different populations as well (*Maeda et al., 2010*; *Kim et al., 2015*; *Klupczynska et al., 2016*; *Proenza et al., 2003*). The reasons for the inconsistent changes in AA levels may be related to the diversity of genetic background, regional differences, nutritional status, and lifestyle factors.
We found three important metabolic pathways which were involved in the metabolism of the altered AAs. Among these, Glu, Asp and Asn were involved in alanine, aspartate and glutamate metabolism. Glu could be converted to $\alpha$-ketoglutarate ($\alpha$-KG) and further entered the tricarboxylic acid cycle (TCA), which is the main way for tumor cells to meet the demands of bioenergy, biosynthesis and redox balance in tumor microenvironment (*Todisco et al., 2019*). Meanwhile, $\alpha$-KG can produce Asp through a series of TCA catalytic reactions in which Asn could also be produced by the catalysis of Asn synthase (ASNS). Therefore, Asp and Asn are also involved in the energy metabolism of the tumor (*Wei et al., 2021*). Likewise, Gly and its precursor Ser participate in TCA through serving as exchange factor of Asn. Notably, ASNS is frequently upregulated in tumors and is associated with poor prognosis of several types of cancer, such as prostate cancer, glioma and neuroblastoma (*Sircar et al., 2012*; *Zhang et al., 2014*). On the other hand, L-asparaginase, an enzyme catalyzing the degradation of Asn, has been widely applied in clinical treatment against multiple types of leukemia and lymphoma (*Haskell & Canellos, 1969*). These evidences suggested a strong dependence of cancer cells for Glu metabolism and energy from TCA.

We noticed that Asp, as well as Orn and Cit, also participated in the Arg biosynthetic pathway. This pathway is a part of the urea cycle, in which intermediates including Orn, Cit, Asp, arginine succinate and Arg are produced by the successive catalysis of carbamyl phosphate synthase 1, arginine succinate synthase 1 and arginine succinate lyase (*Wei et al., 2021*). Although the complete urea cycle works primarily in the liver, some enzymes could relink the intermediates outside the liver. These substances support the survival and proliferation of tumor cells through strong interactions with other metabolic pathways (*Lee et al., 2018*; *Keshet et al., 2018*). Arg is the most consumed AA in the inner necrotic core of tumor mass, indicating its high demand for the survival of tumor cells. Preclinical studies have reported essential roles of Arg in cancer metabolism. For example, Arg starvation caused global transcriptional suppression of metabolic genes including those involved in oxidative phosphorylation, mitochondrial functions, and DNA repair in breast and prostate cancer cells (*Qiu et al., 2014*; *Changou et al., 2014*). Meanwhile, Arg can import into cells, directly activate mTOR pathway, a nutrient-sensing kinase strongly implicated in carcinogenesis. By mTOR mediation, Arg thus has profound impacts on protein synthesis, lipid synthesis and nucleotide synthesis of cancer cells (*Ban et al., 2004*; *Saxton & Sabatini, 2017*). In addition, Arg can also bind G-protein coupled receptor to activate down-stream RAS/ERK or PI3K pathway, another essential pathway in cancer development and progression (*Chen et al., 2021*). Therefore, as the building blocks of Arg biosynthetic pathway, Asp, Orn and Cit might be signaling metabolites and act as transcriptome reprogrammer in cancer metabolism. Besides, as mentioned above, CIT could activate the mTOR1 signaling pathway through modulating the concentration of PA and Rheb activity in the LEL, which also indicating the role of these AAs in cancer signaling transduction (*Bourgoin-Voillard et al., 2016*).

In addition, numerous tumors rely on the availability of extra-cellular Ser to proliferate rapidly (*Jones & Schulze, 2012*). In the conversion process of Ser to Gly, a single-carbon unit could be generated to assist the production of 5, 10-methylene tetrahydrofolate (CH2-THF), while the Gly can also be cleaved by the mitochondrial Gly cleavage system to produce

CH2-THF. CH2-THF could enter the folate cycle for the synthesis of other substances that are involved in cancer metabolism, such as other AAs, nucleotides and lipids (*Tibbetts & Appling, 2010*; *Locasale, 2013*; *Amelio et al., 2014*). Moreover, Ser metabolic pathway probably contribute to NSCLC by serving as a carbon source of nucleotide synthesis and DNA methylation (*Pavlova & Thompson, 2016*). Interestingly, in addition to the de novo-produced Ser, the exogenous donors perhaps affect one-carbon and tumor behavior as well. For instance, a study has found that dietary Ser and Gly deficiency can inhibit the growth of colon cancer cells in mice (*Maddocks et al., 2013*). Besides, Cys might be involved in cancer metabolism through regulating ferroptosis (*Poursaitidis et al., 2017*). Thus, Ser metabolic pathway, as well as Gly and Cys metabolic pathways, may play essential roles in NSCLC metabolism.

The established AA panel in our study was effective in discriminating NSCLC patients from controls, as well as classification of NSCLC patients with different stages and histological types. In the study of *Kim et al. (2015)* and *Shingyoji et al. (2013)*, six AAs (Ser, Gln, Ala, His, Orn, and Lys) were used to establish an index for distinguishing lung cancer cases from healthy controls. Besides the Ser and Orn, we included some other AAs as well, such as Phe, Trp, Ile, Asp, Asn, etc. Although we included more AAs, the AUCs were similar to the results of previous studies. This may be related to sample size, sample composition of clinical stages and pathological types, nutritional status, etc. Notably, a Polish study applied both the Korean and self-built models, and found that the latter has better distinguishing ability (Aal, His, Phe, Cit, Asp, Asn) (*Klupczynska et al., 2016*). Taken together, AAs may have potential value for the classification of NSCLC. However, it may be necessary to establish population-specific AA panels in different countries and regions.

To our knowledge, this is the first case-control study in Chinese population to investigate the alteration of AA levels as well as the involved AA pathways in NSCLC patients, and to evaluate the potential value of AAs in the classification of NSCLC. The enrolled cases and controls were comparable in the basic characteristics. And the approach for determination of AA levels were validated with widely approved standards. However, several limitations should be addressed. Firstly, there are many types of AAs and their derivatives, however, only 22 common ones were selected in our study. More kinds of AAs thus need to be determined and compared in the future. Secondly, AA levels in the human body are affected by a variety of factors including nutritional status, lifestyle factors such as diet, physical activity and other underlying diseases. Although we have matched the cases and controls with gender and age, our results might be still influenced by other potential confounding bias. Therefore, our results still need to be verified in future studies.

## CONCLUSIONS

In conclusion, the current study found that the level of Trp, Phe, Ile, Gly, Ser, Asp, Asn, Cys, Glu, Orn, and Cit were significantly altered in serum of NSCLC patients in comparison with controls. The altered AAs were involved in three important metabolic pathways, including Gly, Ser and Thr metabolism, Ala, Asp and Glu metabolism, and Arg biosynthesis. In addition, this AA panel might pose potential in NSCLC classification.

**Abbreviations**

| | |
|---|---|
| **AA** | amino acid |
| **ENAAs** | essential amino acids |
| **NEAAs** | non-essential amino acids |
| **NSCLC** | non-small lung cancer |
| **Lys** | lysine |
| **Met** | methionine |
| **Ile** | isoleucine |
| **Leu** | leucine |
| **Trp** | tryptophan |
| **Phe** | phenylalanine |
| **Val** | valine |
| **Thr** | threonine |
| **His** | histidine |
| **Asp** | aspartic acid |
| **Asn** | asparagine |
| **Ser** | serine |
| **Glu** | glutamic acid |
| **Gln** | glutamine |
| **Gly** | glycine |
| **Ala** | alanine |
| **Cys** | cystein |
| **Tyr** | tyrosine |
| **Orn** | ornithine |
| **Cit** | citrulline |
| **Arg** | arginine |
| **Pro** | proline |
| **PA** | phosphatidic acid |
| **Rheb** | ras homologue enriched in brain |
| **LEL** | late lysosome/endosome system |
| **ROC** | receiver operating characteristic |
| **AUC** | area under the ROC curve |
| **SCC** | squamous cell carcinoma |
| **TNM** | tumor node metastasis |
| **IQR** | interquartile range |
| **FDR** | false discovery rate |
| **LOD** | limit of detection |
| **LOQ** | limit of quantification |
| $\alpha$**-KG** | $\alpha$-ketoglutarate |
| **TCA** | tricarboxylic acid cycle |
| **ASNS** | asparagine synthase |
| **CH2-THF** | 5, 10-methylene tetrahydrofolate |

## ACKNOWLEDGEMENTS

We wish to thank Xia Luo for assistance in the lab.

### Funding

This study was funded by the National Key Research and Development Plan Program of China (Grant No. 2016YFC1302702) and the National Natural Science Foundation of China (Grant No. 81572071). The funders had no role in study design, data collection and analysis, decision to publish, or preparation of the manuscript.

### Grant Disclosures

The following grant information was disclosed by the authors:
National Key Research and Development Plan Program of China: 2016YFC1302702.
National Natural Science Foundation of China: 81572071.

### Competing Interests

The authors declare there are no competing interests.

### Author Contributions

- Ke Liu conceived and designed the experiments, performed the experiments, analyzed the data, prepared figures and/or tables, authored or reviewed drafts of the paper, and approved the final draft.
- Jiaoyuan Li conceived and designed the experiments, analyzed the data, prepared figures and/or tables, authored or reviewed drafts of the paper, and approved the final draft.
- Tingting Long, Yi Wang, Tongxin Yin and Jieyi Long analyzed the data, authored or reviewed drafts of the paper, investigation, and approved the final draft.
- Ying Shen conceived and designed the experiments, performed the experiments, authored or reviewed drafts of the paper, and approved the final draft.
- Liming Cheng conceived and designed the experiments, authored or reviewed drafts of the paper, and approved the final draft.

### Human Ethics

The following information was supplied relating to ethical approvals (i.e., approving body and any reference numbers):

Institutional Review Board of Tongji Medical College, Huazhong University of Science and Technology

### Data Availability

The original measurements are available in the Supplementary File.

### Supplemental Information

Supplemental information for this article can be found online at http://dx.doi.org/10.7717/peerj.13272#supplemental-information.

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
