# Peer review of "Changes in serum amino acid levels in non-small cell lung cancer: a case-control study in Chinese population"

_PeerJ, doi:10.7717/peerj.13272_

## Round 0.1 · original submission · Major Revisions

Dear authors,

First of all, we regret the long time spend for the first decision in your Ms. It was really difficult to find adequate reviewers for your work. Thank you for your patient.

We have now the opinion of two Reviewers in your work. As you will see both reviewers found your paper interesting and with novelty. However, both consider that Major modifications are necessary before considering the manuscript further.

Please revise your manuscript and answer all the concerns of the Reviewers.

Reviewer 1 ·

Basic reporting

In this study, the authors evidenced that the level of several amino acids were modified in Chinese NSCLC patients compared to control patients. One amino acid was also expressed differentially depending on the subtype of NSCLC (ADC or SSC). While the alteration of AAs in serum is well established, no study reported such effects in a specific population such as Chinese patients and never by considering subgroups of ADC and SSC. This study is interesting and the article is well written and respects the standards for an original research article. Nevertheless, several aspects mentioned in the comments below should be clarified to consider this manuscript for publication

Experimental design

In this study, the authors quantified the level of amino acids in serum to evidence that the level of several amino acids was modified in Chinese NSCLC patients compared to control patients and that one amino acid was also expressed differentially depending on the subtype of NSCLC (ADC or SSC). While the method used is well appropriate for such aim (in term of number of patients included in the cohort, accuracy of the quantification method, statistical approach) and several important details are well described, it would be helpful to edit the manuscript according to the comments below
- line 136, the authors justify the choice of using Kolmogorov–Smirnov test rather than other statistical multivariate tests
- line 141: “a validation set with a ratio of 7:3.” The authors should specify what is this ratio
- Line 178” After adjustment, the concentration of Trp was significantly decreased,” The authors should explain what are these adjustments.
- Line 182-190: The authors consider only the P-FDR lower than 0.01, but the ratio of AA level was pretty close to 1 (i.e, 1.07). Is it relevant to consider AAs with such ratio as significantly deregulated? In my point of view, it would be more correct to apply a threshold a ratio from which the AAs is considered significantly deregulated with a P-FDR lower than 0.01
- Line 247: It is well known that the nutrition impacts the AA expression levels. So, I was wondering the patients followed a specific diet or at least if those data where recorded to ensure that the difference of expression is link to NSCLC and to different diets.

Validity of the findings

Through this study, the authors evidenced modulations of several AA expression level in serum of Chinesse patients to discriminate the NSCLC patients to control patients. They also evidence that Asp expression change depending on the subtype of the NSCLC (ADC or SSC) It will be helpful to classify teh control from NCSCL cancer,but the auhtors should answer the comments below to improve the manuscript

- Table 1. I have concerns that the difference between control and cases groups is significant regarding smoking and BMI aspects. This may have impacts in AA expression level and interferes in the results of the study.
- The authors should present a table of clinical characteristic for the classification of subgroups as they did in table 1
- the authors should discuss about the interest of their method campared to the methods that are currently used in clinics

Additional comments

1) An abbreviation list would be helpful to read the article such as for EAA, NEAA, IQR
2) Line 68: The authors should also add that several AAs such as CIT also activate the mTOR1 signalling pathway through the modulation of the concentration of PA and Rheb activity in the LEL by citing, for instance, the review S. Bourgoin-Voillard et al Proteomics 2016, 16, 831–846
3) Line 187-190: “However, except for the higher Asp in SCC (P-FDR<0.01), there was no significant difference in AA levels between adenocarcinoma and SCC.” The comparison of AA expression level in subgroups is not shown for all AAs presented in the article. It would be interesting to present it to all AAs and not on a few of them. Otherwise the authors should justify why they selected those AAs.
4) Table 2: As CIT seems deregulated, it should be in bold
5) Line 249-262: the authors should also discuss about the role of CIT in mTOR signaling pathway activation
6) It would be interesting to establish a ROC curve by considering not only one AAs, but several AAs. Such combination may increase the specificity of the biomarker test

·

Basic reporting

This paper is intriguing and well-written, offering insights into the topic of cancer. I only have two small points to make.

Experimental design

Please provide the internal standard utilized to determine AA.
Please provide further information for metabolic pathway analysis.
Please provide a power analysis of your study as well as type-II error statistics.

Validity of the findings

Please provide a power analysis of your study as well as type-II error statistics.

---

## Round 0.2 · accepted · Accept

Dear Authors

Thank you for attending to the reviewers' concerns in a proper way.

Your manuscript is now acceptable for publication in PeerJ
Congratulations

Reviewer 1 ·

Basic reporting

The authors improved properly the manuscript according to my previous comments. The manuscript is thus, ready for publication.

Experimental design

no comment

Validity of the findings

no comment

·

Basic reporting

The manuscript is well written and ready for publication.

Experimental design

Experimental methods are well designed.

Validity of the findings

The findings are validated using training data.